# Genome-Wide Analysis Identifies Candidate Genes Encoding Feather Color in Ducks

**DOI:** 10.3390/genes13071249

**Published:** 2022-07-14

**Authors:** Qixin Guo, Yong Jiang, Zhixiu Wang, Yulin Bi, Guohong Chen, Hao Bai, Guobin Chang

**Affiliations:** 1College of Animal Science and Technology, Yangzhou University, Yangzhou 225009, China; dx120190114@yzu.edu.cn (Q.G.); jiangyong@yzu.edu.cn (Y.J.); wangzx@yzu.edu.cn (Z.W.); ylbi@yzu.edu.cn (Y.B.); ghchen2019@yzu.edu.cn (G.C.); 2Joint International Research Laboratory of Agriculture and Agri-Product Safety, The Ministry of Education of China, Yangzhou University, Yangzhou 225009, China

**Keywords:** duck, plumage color, GWAS, selective sweep, melanin

## Abstract

Comparative population genomics and genome-wide association studies (GWAS) offer opportunities to discover human-driven detectable signatures within the genome. From the point of view of evolutionary biology, the identification of genes associated with the domestication of traits is of interest for the elucidation of the selection of these traits. To this end, an F_2_ population of ducks, consisting of 275 ducks, was genotyped using a whole genome re-sequence containing 12.6 Mb single nucleotide polymorphisms (SNPs) and four plumage colors. GWAS was used to identify the candidate and potential SNPs of four plumage colors in ducks (white, spot, grey, and black plumage). In addition, FST and genetic diversity (π ratio) were used to screen signals of the selective sweep, which relate to the four plumage colors. Major genomic regions associated with white, spotted, and black feathers overlapped with their candidate selection regions, whereas no such overlap was observed with grey plumage. In addition, *MITF* and *EDNRB2* are functional candidate genes that contribute to white and black plumage due to their indirect involvement in the melanogenesis pathway. This study provides new insights into the genetic factors that may influence the diversity of plumage color.

## 1. Introduction

Coloration plays a critical evolutionary role in both natural and sexual selection. A recent feature color study suggests that the derived traits produced during animal domestication tend to undergo predictable sequential changes [1,2]. In the early phases of domestication, traits related to color are domesticated first in the earliest stages of domestication [3,4,5,6,7], followed by traits affecting food ability, changes in coat or feathers and structure, and finally, differences in behavior. The color of a duck’s plumage is one of the simplest and most straightforward derived economic traits. [8]. The color of the plumage is a favorable trait that meets the need for various types of downs as stuffing for jackets or blankets and makes the carcass simple to clean [9]. Color production is a complex process regulated by several factors, mainly genetics, epigenetics, cytology, physiology, and environmental factors. However, while genetic factors are the main influence on color production, the functions of important genes are still poorly understood. Understanding the relationship between color and genetics is important for understanding the complex coloration mechanisms of duck feathers.

Previous studies on duck feather color have been limited to investigations of candidate loci such as melanin-related genes. Many studies have reported an association between phenotypic and genomic variation for many candidate loci, particularly the melanocortin-1 receptor (*MC1R*) [10,11,12,13,14,15], microphthalmia-associated transcription factor (*MITF*) [16,17], endothelin receptor B (*EDNRB*) [18,19], tyrosinase (*TYR*), and tyrosine 1 (*TYRP1*) [20,21,22]. In bananaquit (*Coereba flaveola*), single point mutations in *MC1R* results in two different forms, which represent the association between plumage and gene variation [23]. Meanwhile, the investigations of *MC1R* in other bird species also suggested that *MC1R* has a similar association with plumage color [24,25]. However, studies related to duck feather color showed that the deletion of the *MITF* promoter region affected the feather color of Peking ducks. Thus, in several vertebrate species, the *MITF* gene is an important developmental locus with complex regulatory roles involved in pigmentation and melanocyte development [26,27,28,29,30,31]. The other key genes, such as *TYR*, *TYR1*, and *DCT*, involved in the melanogenesis pathway also regulate plumage color. In addition to the identification of SNP, copy number variants (CNV) and insertion/deletion (indel) from across the genome are associated with black or spot plumage. Previous research show that the *EDN3* gene is associated with dark pigmentation in two local chicken breeds [32]. Genome-wide association study (GWAS) using black and non-black chickens and *SHH* and *NUAK* genes showed a strong association [33]. Although these findings are a step toward the epistatic causation of the genes of interest in plumage color at different developmental stages in chickens, they also confirm that further studies are essential to elucidate the molecular mechanisms underlying the genetics of plumage color in poultry. However, the potential molecular mechanisms of plumage color in different species or breeds differ. Thus, we conducted a GWAS and a genome-wide selective sweep scan to identify genomic regions that may explain the phenotypic differences observed between white, spot, and black plumage in the present study. These studies provide insights into the molecular regulatory mechanisms of melanin deposition in duck feathers and genetic improvement of duck plumage color.

## 2. Materials and Methods

### 2.1. Ethical Approval

The Ministry of Science and Technology (Beijing, China) released Regulations on the Administration of Experimental Animals in 1988, and all research of ducks was conducted in compliance with those regulations (last modified in 2001). The Yangzhou University’s Animal Care and Use Committee authorized the experimental protocols (YZUDWSY2017-11-07). Every attempt was made to reduce discomfort and suffering in the animals.

### 2.2. Samples and Sequencing

The F2 resource population, which crosses the Chinese crest duck (CC duck) and cherry valley duck (CV duck), was obtained from the Laboratory of Poultry Genetic Resources Evaluation and Germplasm Utilization at Yangzhou University. The ducks were raised in stair-step cages under the recommended environmental and nutritional conditions at the conservation farm of the Ecolovo Group, China. The CC ducks with a black shank and beak and white plumage are an indigenous Chinese breed. The CV ducks with a yellow shank and beak and white plumage are a commercial breed. In the F_1_ generation, thirty CC ducks and six CV ducks were randomly selected and divided into six families. To produce F2 offspring, 30 male ducks were crossed with 150 unrelated females. A total of 275 ducks were used in the next experiment. To identify candidate genes associated with plumage, we classified plumage colors into white, spotted, grey, and black. Blood samples were used to collect high-quality DNA at 42 days of age.

Genomic DNA (gDNA) was extracted from blood samples by DNA extraction using a kit (QIAampR DNA Blood Mini Kit, (QIAGEN, Valencia, Santa Clarita, CA, USA)), following the manufacturer’s protocol. According to the Illumina protocol, two paired-end sequencing libraries with insert sizes of 350 bp were constructed (Illumina, San Diego, CA, USA). The Illumina NovaSeq platform was used for the sequencing of all libraries.

### 2.3. Variants Calling and Genotyping

Burrows–Wheeler Aligner (BWA, version: 0.7.17-r1188, Heng Li, MA, USA) [34] software was used to align all clean reads from 275 samples to the CC duck genome (unpublished) (settings: mem −t 4 −k 32 −M −R). Variant calling was carried out for all samples using the UnifiedGenotyper method from Genome Analysis Toolkit (version 3.7). SNPs were filtered using the snpfilter.pl Perl script, which can be found at https://github.com/genome/somatic-sniper/tree/master/src/scripts (15 May 2022). A minimum minor allele frequency (MAF) of 5%, a genotype call rate of 95% or higher, and Hardy–Weinberg equilibrium were required for SNP selection (*p* > 0.00001). 12.6 Mb of SNPs were chosen for GWAS after filtering.

### 2.4. Population Structure Analysis

Multidimensional scaling (MDS) was used to evaluate population structure by PLINK 1.9 [35]. Using the independent-pairwise option, a 1000 bp window, five steps, and a r2 threshold of 0.2 independent SNPs were found on all autosomes. These independent SNP markers were used to construct pairwise identity-by-state (IBS) distances between all individuals, and MDS components were acquired using the mds-plot option based on the IBS matrix. These distinct SNP markers were used to create a relative kinship matrix.

### 2.5. Genome-Wide Association Analysis

The GWAS analysis for plumage color used the linear mixed model by EMMAX (Effective Mixed Model Association Expedited, version: beta, Hyun Min Kang, Michigan, USA) [36] software based on case–control methods. When estimating the model variance components to take into account population structure, EMMAX simplifies and saves the assumption that the effect of each individual SNP on the trait is typically small. EMMAX uses the REML model to estimate the variance components.
y=Xa+Zb+e
where Z is the genotype value of the candidate SNP, b is the regression coefficient of the candidate SNP, e is the random residual, y is a vector of plumage colors, X is the incidence matrix for a random additive effect, and a is the column vector of random additive effects. The additive variance (*a*^2^), the variance of random residuals (*e*^2^), the identity matrix (I), the IBS kinship matrix (K), and the additive variance (*a*^2^) make up the phenotypic variance–covariance matrix ($Var\left(y\right)=Var\left(a\right)+Var\left(e\right)=Ka^{2}+Ie^{2}$. The Bonferroni correction was used to determine the significance level (0.05/total number of SNPs) for the association of SNP and CNV markers with various characteristics.

### 2.6. Detection of Selection Signatures within Populations

The fixation index (F_ST_), which gives insight on genome-wide difference among different groups, was used to determine the degree of population differentiation between ducks with black and white plumage. As a result, we used vcftools [37] to construct population fixation statistics (F_ST_) with a sliding window and step size of 2 kb and 1 kb, respectively (—fst-window-size 2000—fst-window-step 1000). The value at the whole-genome level among the different groups was taken to be the weighted F_ST_ of all sliding windows. A measure of a population’s degree of variability is called nucleotide diversity (π). Using genotypes from each group independently, nucleotide diversity was estimated using vcftools with a 1 kb sliding window. Furthermore, genetic diversity ratios between different groups were calculated to estimate plumage color-related regions. Log2 (π_other_/π_white_) was used as an estimate of white plumage-related signals and log2 (π_other_/π_black_) was used as an estimate of black plumage-related signals. The top 1% was used as the significance cut-off value. The candidate selective sweeps discovered using the above-mentioned approaches (F_ST_ and log2 (θπ)) were annotated using the Variant Effect Predictor (VEP) tool [38,39].

### 2.7. Gene Ontology (GO) and Kyoto Encyclopaedia of Genes and Genomes (KEGG) Analyses

The annotation of associated genes in a specific region upstream and downstream of the actual locations of the relevant SNPs was carried out based on the LD attenuation distance determined by PopLDdecay [40]. From the mallard genome, the required gene sequences were retrieved, translated into a protein sequence, and then submitted to KOBAS 3.0 [41]. The hypergeometric test and Fisher’s exact test were employed as the statistical techniques, and chicken was chosen as the reference species.

## 3. Results

### 3.1. Population Structure, Linkage Disequilibrium, and Snp Distribution

A total of 12,201,978 SNPs with minor allele frequency >0.05 and maximum missing rate 0.1 were found among the 275 ducks tested and used in the following analyses. According to the results of the MDS plot, there was no obvious grouping in the F_2_ segregated population. Individuals with different plumage patterns were evenly distributed among the three clusters (Figure 1a). Nevertheless, the examined phenotypes were not significantly affected by this stratification because individuals with various features were equally distributed across the two clusters. Additionally, the half-maximum LD’s LD was 0.273 and the highest LD was 0.572. At distances of 82,142 the relevant LD threshold was established at r2 = 0.1 (Figure 1b). All filtered SNPs were dispersed among the 37 autosomal chromosomes, chromosome Z, and a few unplaced contigs (Unplaced), with an average density of 11,259.45 SNPs/Mb (Figure 2).

### 3.2. Genome-Wide Association Study Identified the Candidate Variants of Plumage Color

EMMAX software was used to conduct the genome-wide association analysis in the present study. The Q–Q plot illustrated that the model used in this study for GWAS analysis was reasonable. The lambda (inflation factor (λ)) of black, grey, spot, and white plumage were 0.96, 0.98, 0.95, and 0.86, respectively (Figure 3), and the points at the upper right corner of the plots are the significant markers associated with the traits under study. Thus, population stratification was adequately controlled.

The Manhattan plot of plumage color showed that 13 significant SNPs associated with black plumage traits were identified using the threshold of suggestive and significant *p*-values (threshold = 0.05/total number of all SNPs = 3.94885 × 10^−9^), most of which were located on chromosome 14 (ALP 14) (12 SNPs) and ALP 19 (1 SNP) (Figure 4). According to the LD decay analysis, there were significant SNPs in *MAP4K4*, *NKAPL*, *TMLHE*, *SPRY3*, *BRN3*, *EDNRB2*, *VAMP7*, *SCEL*, and *TSPAN10* (Appendix A). For white plumage, a total of 8857 (threshold = 3.94885 × 10^−9^) and 7377 (threshold = 7.8977 × 10^−10^) SNPs were significantly associated with white plumage (Figure 4). Based on LD decay, *SYNPR*, *CADPS*, *PTPRG*, *FHIT*, *QTSA-12457*, *TRAPPC10*, *ACOX2*, *PRKAR2A*, *SLC25A20*, *USP19*, *SLC6A6*, *CHST13*, *TXNRD3*, *PLXNA1*, *CHCHD6*, *ABTB1*, *RPN1*, *RAB7A*, *CRBN*, *PSMD6*, *FAM3D*, *MGLL*, *H1FX*, *GXYLT2*, *VAR1*, *FABG*, *MITF*, and the other 30 genes were nearest to the significant SNPs (threshold = 7.8977 × 10^−10^), which were associated with white plumage. The results of the candidate gene enrichment analysis, melanogenesis pathway, MAPK/ERK pathway, etc. (Appendix A). A total of 66 SNPs (threshold = 3.94885 × 10^−9^) were located at ALP 11 and nearest the genes *CNTN6*, *CHL1*, *EIF4E3*, *MITF*, *FRMD4B*, *ARL6IP5*, *FAM19A1*, *SUCLG2*, *CNTN3*, *FOXP1*, and *SLC16A7* (Figure 4). In addition, a total of eight and twelve SNPs located on ALP14 and ALP30 (threshold = 1 × 10^−6^), respectively, were potentially associated with spot plumage (Figure 4 and Appendix A). These genes were closest to *SPRY3*, *BRN3*, *NSDHL*, *EDNRB2*, *CETN2*, *ANKRD52*, *PRPH*, *ADIPOQ*, *DGKA*, *PA2G4*, *ESYT1*, *TIMELESS*, *TMEM106C*, and *DNAJC22*. For grey plumage, 18 potential SNPs were associated with grey plumage. *APC2*, *GFPT2*, *CDH19*, *BTK*, *GLA*, *PLP1*, *RAB9B*, *TMLHE*, *IFT122*, *CORO2B*, *SMAD3*, *ENDOD1*, *CNOT6*, *MC2R*, *PRPS1*, *SPRY3*, *SMAD6*, and *CWC15* were the closest SNPs (Appendix A). To determine the relationship between the four different color plumage phenotypes, a Venn analysis of significant SNPs of each color was performed. The Venn results showed that the 66 significant SNPs nearest to the genes *CNTN6*, *CHL1*, *EIF4E3*, *MITF*, *FRMD4B*, *ARL6IP5*, *FAM19A1*, *SUCLG2*, *CNTN3*, *FOXP1*, and *SLC16A7* overlapped between spot and white plumage (Figure 5).

### 3.3. Selective Sweeps Analysis of the Candidate Genes of Color Plumage

To identify the candidate genes for color plumage, we scanned the genome for regions with extreme divergence in allele frequency (F_ST_) and the highest differences in genetic diversity (log2 π ratio) in 2-kb sliding windows and 1-kb steps in four color plumage phenotypes to detect candidate divergent regions (CDRs). In total, we identified 10 white plumage-related CDRs (F_ST_ ≥ 0.14 and log2 π ratio > 1.45) (Figure 6a and Appendix A). During annotation of CDRs, we found that these CDRs are located on chromosome 11. These regions harbored four genes (*MITF*, *FRMD4B*, *FAM19A1*, and *SUCLG2*). Among these, *MITF* plays a crucial role in melanogenesis. In addition, combined with the GWAS results, these genes were also significantly associated with white plumage (Figure 6a). For spot plumage, we identified 34 spot plumage-related CDRs (F_ST_ ≥ 0.04 and log2 π ratio > 0.85) located on chromosomes 1 (1 CDR), 11 (32 CDRs) and 14 (1 CDR) (Appendix A). Moreover, these spot plumage-related CDRs harbor eight genes, including *ATXN7*, *THOC7*, *SPRY3*, *MITF*, *LGSN*, *SNTN*, *POU4F3*, and *SUN3* genes. Combined with the results of the spot plumage GWAS, the *MITF* was also a significant associate with spot plumage.

Furthermore, we identified 166 grey plumage-related CDRs (F_ST_ ≥ 0.07 and log2 π ratio > 4.48) located on chromosomes 1 (57 CDRs), 2 (61 CDRs), 3 (13 CDRs), 5 (10 CDRs), 13 (4 CDRs), 15 (3 CDRs), 16 (2 CDRs), 22 (11 CDRs), and Z (5 CDRs) (Appendix A). However, we did not find any significant SNPs located in these CDRs. Meanwhile, the FST values of the spot and grey plumage were extremely low. We speculate that this may be due to the fact that speckle and grey plumage are complex traits and due to the consistency of the genetic background of the F2 population. Finally, we identified 26 CDRs related to black plumage located on chromosomes 11 and 14 (Appendix A). These black plumage-related CDRs harbor 13 genes, including *KLF15*, *SLITRK2*, *KIAA1210*, *ZKSCAN4*, *AIFM1*, *MARS2*, *SPRY3*, *POU4F3*, *CHIC1*, *FMR1*, *EDNRB2*, *FLT1*, and *FIP1L1*. Combined with the results of the black plumage GWAS, *EDNRB2* was also significantly associated with black plumage (Figure 6b).

## 4. Discussion

Feather color plays an important role in bird domestication [42,43]. Bird feather color is an important economic trait, and the value of filled down products varies significantly according to the color of bird feathers [44]. For plumage color of birds, pigments and structure are the two main and important sources [45,46]. Melanin is the main type of pigment that may control the color of plumage, skin, etc. Several articles have recently reported candidate loci for bird feather coloration. A previous study on Swainson’s thrush (*Catharus ustulatus*) confirmed that relevant regions, including *TYRP1*, were selected during evolution [47]. The findings related to chicken feather color confirmed 18 potential candidate regions, including *HNF4BETA*, *CKMT1B*, *TBC1D22A*, *RPL8*, *CACNA2D1*, *FZD4*, *SGMS1*, *IRF8*, *OPTN*, *LOC420362*, *TRABD*, *OVODA1*, *DAD1*, *USP6*, *RBM12B*, miR-1772, miR-1709, and miR-6696; 89 gene–gene combinations that may lead to changes in feather color may be primarily responsible for chicken feather color [48]. Additionally, the diversity of *MC1R* was associated with plumage color in pigeons [49]. Studies on ducks have confirmed that some *MITF* mutations are associated with white feathers [8,9,50,51]. In the present study, it was also confirmed that *MITF* mutations were significantly associated with white feathers using GWAS and selection sweep analysis. A important regulator of melanocyte migration and differentiation is thought to be the MITF protein [52]. Additionally, it has been suggested that POU members control the transcription of the MITF gene. The correlation study between allele frequencies and phenotypes, which implied a coordinating influence of these two loci on melanin deposition in ducks, was also supportive of the regulatory linkages between POU members and the MITF gene [53]. Furthermore, our data also indicate that *POU4F3* is associated with spot and black plumage. We speculate that *MITF* and *POU4F3* likely have a synergistic effect on the regulation of melanin synthesis and that their mutations contribute to phenotypic differences in plumage melanin deposition among individuals. GWAS analysis showed that *MC2R* was significantly associated with grey plumage. *MC2R* is also a member of the melanocortin receptor (MCRs) family. *ACTHR* is also known as *MC2R*. *ACTHR* is unique among MCRs because it binds to one sole ligand, *ACTH*, making it a very attractive research object for molecular pharmacologists. *ACTH* promotes melanin production [54]. Thus, MC2R mediates *ACTH* to promote melanin deposition, which leads to the formation of grey feathers; however, the specific mechanism requires further experimental verification. Other studies have also reported that the *EDNRB2* is a candidate gene associated with black color in live animals and poultry [55,56]. *EDNRB2* is a key gene involved in melanogenesis. Our data also indicated that *EDNRB2* is significantly associated with black plumage.

## 5. Conclusions

This study identified candidate genes strongly associated with plumage color in ducks. *MITF* and *EDNRB2* were significantly associated with white and black plumage. In addition, *POU4F3* and *MC2R* may synergistically regulate melanin deposition, which affects feather color. These findings confirm the intricacy of the mechanisms governing this feature and shed light on the genetic basis of ducks’ plumage color. Additional research is necessary to refine the presented results and further investigate the molecular mechanisms underlying plumage color.

## Figures and Tables

**Figure 1 genes-13-01249-f001:**
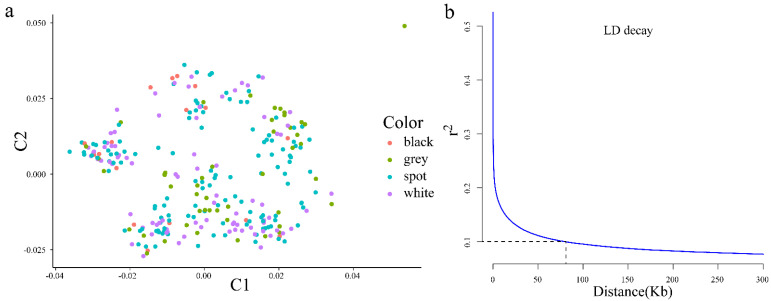
MDS plot (**a**) and LD decay plot (**b**) of F2 population by all SNPs.

**Figure 2 genes-13-01249-f002:**
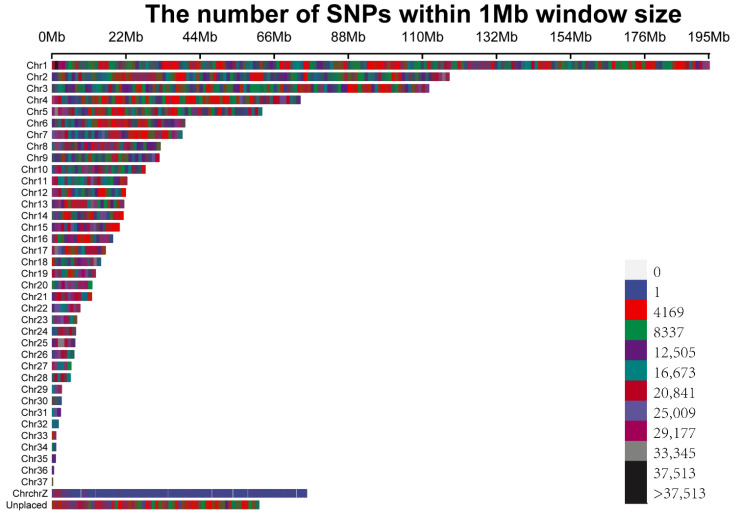
SNP density plot.

**Figure 3 genes-13-01249-f003:**
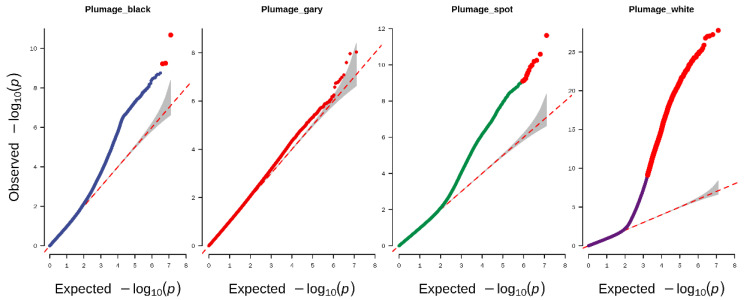
Quantile–quantile (Q–Q) plots from GWAS for plumage color trait in duck. The red lines indicate the null hypothesis of no true association. The red tail means the significant associate SNPs.

**Figure 4 genes-13-01249-f004:**
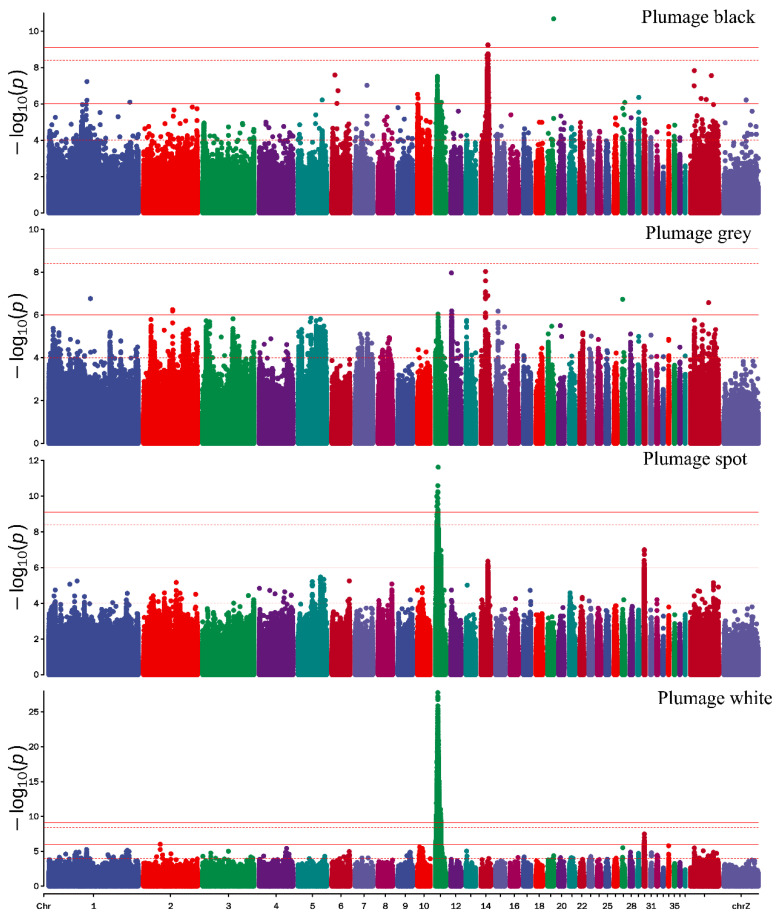
Manhattan plot of four different color plumage phenotypes. The lines from top to bottom in the Manhattan plot represent thresholds of 7.897699 × 10^−10^, 3.948849×10^−9^, 1 × 10^−6^ and 1 × 10^−4^, respectively.

**Figure 5 genes-13-01249-f005:**
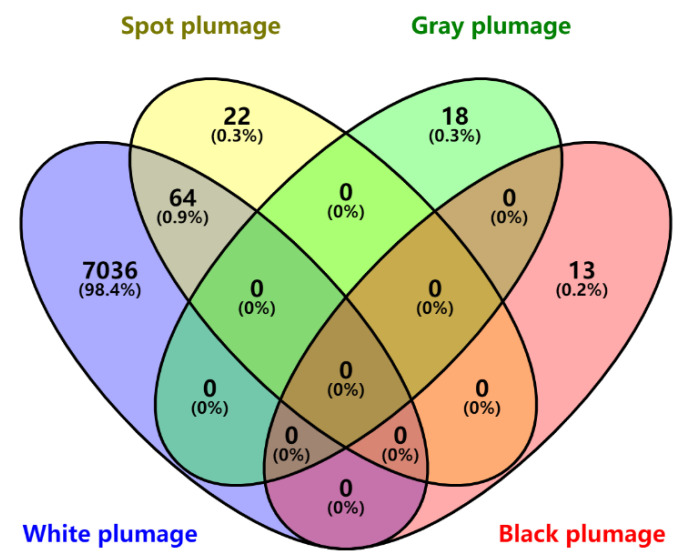
Venn analysis of significant and potential SNPs of four-color plumage phenotypes.

**Figure 6 genes-13-01249-f006:**
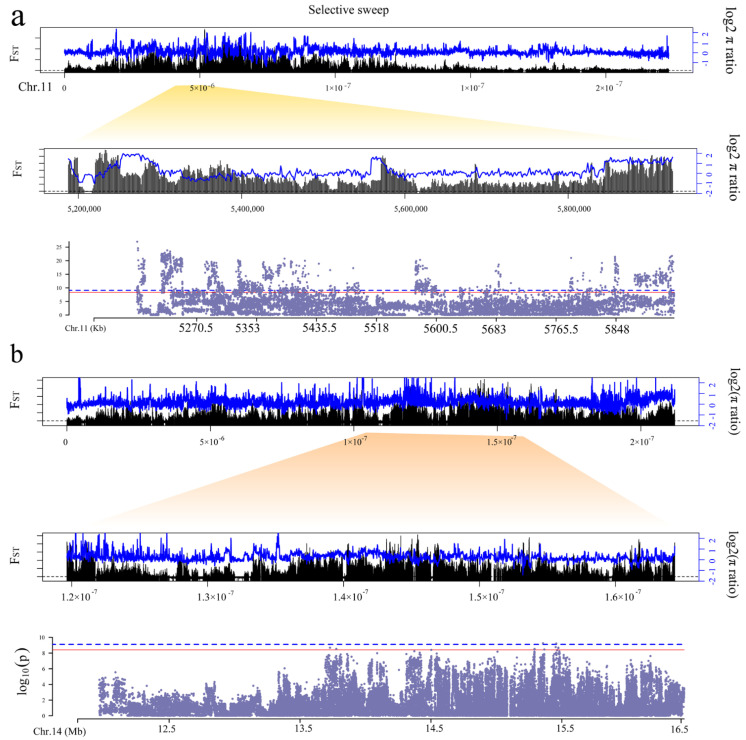
Signatures of selection of white (**a**) and black (**b**) plumage in duck. The blue dashed line represents a threshold of 7.897699 × 10^−10^ and the red solid line represents a threshold of 3.948849 × 10^−9^.

## Data Availability

The genome assembly and all re-sequencing data used in this research were deposited in the Genome Sequence Archive (GSA) at the National Genomics Data Center (http://bigd.big.ac.cn/) Beijing Institute of Genomics, Chinese Academy of Sciences (GSA: CRA005019).

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
