# Peer review of "Genome-Wide Analysis Identifies Candidate Genes Encoding Feather Color in Ducks"

_genes, 2022, doi:10.3390/genes13071249_

Round 1

Reviewer 1 Report

- L26: The abstract must be needed by clear recommendation and informative conclusion.

- L28: Increase the keywords.

- Add a phrase to the introduction to highlight the study hypothesis.

- I suggest extensive English editing in a certified office.

- What about statistics?

- Update the discussion with recent citations up to 2022.

Author Response

- L26: The abstract must be needed by clear recommendation and informative conclusion.

Response: Thanks for your kind suggestions. We have the clear recommendation and informative conclusion in abstract.

- L28: Increase the keywords.

Response: Thanks for your kind suggestions. We have increased the keywords.

- Add a phrase to the introduction to highlight the study hypothesis.

Response: Thanks for your kind suggestions. We have provided the highlight the study hypothesis.

- I suggest extensive English editing in a certified office.

Response: Thanks for your kind suggestions. We have embellished the manuscript.

- What about statistics?

Response: Thanks for your comment. We have added the statistics to manuscript.

- Update the discussion with recent citations up to 2022.

Response: Thanks for your comment. We have updated the reference of discussion to 2022.

Reviewer 2 Report

The manuscript aims to contribute to the dissection of genetic basis of feather color in ducks. Authors resequenced 275 indiviuals from an f2 population and were able to suggest candidates by combining gwas, FST and genetci diversity analysis.
In general, the manuscript is interesting and methods applied are adequate.
Some additional considerations are detailed.

INTRODUCTION
L32. Review the sentence.

MATERIAL AND METHODS
L86. white plumage?
L86-87. Review the sentence "In F1 generation was hybrid by 30 CC ducks and 6 CV ducks were randomly selected to be divided into six families.".
L89-90. It should be interesting if authors could provide examples of the phenotypes analysed (maybe a supplementary material).
L98. 275 sequences or samples? Why the reference genome was an unpublished genome?
L.102 It should be interestinf is authors provide more details about the perl script used to filter the SNPS. Is it part of the GATK or an in house script?
L.113-114 Analyses were performed considering case-control?
L235. Based?
RESULT
Figure 2A. Different phenotypes should be plotted in different colors to facilitate the interpretation.
Figure 4. Plumage_gary?

DISCUSSION
L357. were association?
L359. Review the sentence "In present work, In the present study, it was also confirmed"
CONCLUSIONS

Author Response

The manuscript aims to contribute to the dissection of genetic basis of feather color in ducks. Authors resequenced 275 indiviuals from an f2 population and were able to suggest candidates by combining gwas, FST and genetci diversity analysis.

In general, the manuscript is interesting and methods applied are adequate.

Some additional considerations are detailed.

Response: Thanks for your kind suggestions. We’ve revised the manuscript and made a point-by-point response to your comments. Please see below for details.

INTRODUCTION

L32. Review the sentence.

Response: Thanks for your comment. We have changed this sentence.

MATERIAL AND METHODS

L86. white plumage?

Response: Thanks for your comment. We have changed this mistake.

L86-87. Review the sentence "In F1 generation was hybrid by 30 CC ducks and 6 CV ducks were randomly selected to be divided into six families.".

Response: Thanks for your comment. We have changed the sentence “In F1 generation was hybrid by 30 CC ducks and 6 CV ducks were randomly selected to be divided into six families.” to “To produce F2 offspring, 30 male ducks were crossed with 150 unrelated females.”

L89-90. It should be interesting if authors could provide examples of the phenotypes analysed (maybe a supplementary material).

Response: Thanks for your comment. We have provide the examples of the phenotype in supplementary material.

L98. 275 sequences or samples? Why the reference genome was an unpublished genome?

Response: Thanks for your comment. We have changed this mistake. The reference genome used in the current study was assembled using Chinese crested duck as a sample, and although it has not been published for the time being, the related article is in submission status and the article is preprinted in bioRxiv.

The references are as follows:

Chang, G., et al., The first crested duck genome reveals clues to genetic compensation and crest cushion formation. bioRxiv, 2021: p. 2021.07.25.452189.

L.102 It should be interestinf is authors provide more details about the perl script used to filter the SNPS. Is it part of the GATK or an in house script?

Response: Thanks for your comment. The perl script were performed to filter the SNPs by quality, depth and remove the indels from the vcf file. And the link of perl script were provide in manuscript. 

L.113-114 Analyses were performed considering case-control?

Response: Thanks for your comment. GWAS analysis is done by a case-control strategy.

L235. Based?

Response: Thanks for your comment. We have changed this sentence.

RESULT

Figure 2A. Different phenotypes should be plotted in different colors to facilitate the interpretation.

Response: Thanks for your comment. We have changed the figure 2a.

Figure 4. Plumage_gary?

Response: Thanks for your remind. And we have changed this mistake.

DISCUSSION

L357. were association?

Response: Thanks for your comment. And we have changed this mistake

L359. Review the sentence "In present work, In the present study, it was also confirmed"

Response: Thanks for your comment. We have changed this sentence.

Round 2

Reviewer 1 Report

Accept